# Estimation and Development-Potential Analysis of Regional Housing in Ningbo City Based on High-Resolution Stereo Remote Sensing

Xiao Du [1], Li Wang [1,*], Feng Tang [1], Shiguang Xu [1], Shakir Muhammad [2], Biswajit Nath [3] and Zheng Niu [1,4]

1   State Key Laboratory of Remote Sensing Science, Aerospace Information Research Institute, Chinese Academy of Sciences, Beijing 100101, China
2   Department of Space Science, Institute of Space Technology, Islamabad 44000, Pakistan
3   Department of Geography and Environmental Studies, Faculty of Biological Sciences, University of Chittagong, Chittagong 4331, Bangladesh
4   University of Chinese Academy of Sciences, Beijing 100049, China
*   Correspondence: wangli@radi.ac.cn

**Abstract:** With the challenges brought about by the COVID-19 pandemic, China's real-estate market has been facing new bottlenecks. The solution lies in an in-depth understanding of regional real-estate conditions. In the study of housing, remote sensing technology can help to extract building height as well as to calculate the number of floors and estimate the total amount of housing. It is more efficient and accurate compared to conventional statistical and sampling methods. Remote sensing is widely used in real-estate research and building height estimation, whereas it is less frequently used for the total estimation of urban housing. In this context, we used Chinese satellite GF-7 stereopair images, point of interest (POI) data, and other data combined with the digital surface model (DSM) and shadow methods to calculate the height of residential buildings. An efficient and accurate method system was then established for estimating the total housing and per capita living area (PCLA). According to the calculation of the PCLA of each district in Ningbo City (China), it was found that different regions were suitable for different development paths. Based on this, the driving factor model was derived and the real-estate development potential of Ningbo city was quantitatively analyzed. The results showed that Ningbo City, a first-tier city with a large population inflow, still has potential for real-estate development.

**Keywords:** high resolution; GF-7; real estate; development potential; buying drivers; Ningbo city

## 1. Introduction

Housing problems affect people's livelihood and social development. They impact economic development, social harmony, and stability. In the past two decades, China's real-estate market (REM) has gone through various stages of development, including an early start, rapid development, policy coordination, and finally, stabilization. The overall REM has shown a downward trend since 2022 and real-estate investment has also declined. In 2022, sales area, volumes of commercial housing, and real-estate investment across the country decreased by 24.3%, 26.7%, and 10.0%, respectively [1]. As the pillar industry is linked to dozens of upstream and downstream industries, real estate plays a major role in China's economic development. However, the current situation of the REM in Chinese cities is very different. If we have a deep understanding of the local real-estate situation, we can make relevant arrangements according to local conditions, mastering the basic situation of housing in first-tier cities, exploring the potential of real-estate development as a reference for formulating policies, and ultimately promote the stable and healthy development of the REM. Therefore, it is important to solve all the associated problems.

A statistical study focusing on total housing is of great significance during the investigation of housing issues to understand the development of real estate, improve the

supply of housing, and formulate objectives and scientific goals for real-estate development. However, China's housing planning started late and some problems, such as difficulty in obtaining basic housing data, still exist in the country. The total amount of housing and the per capita living area (PCLA) are usually obtained using statistics and sampling techniques [2]. This method of data acquisition still has many shortcomings, such as poor sample survey accuracy, lag in data statistics, limited housing properties, insufficient statistics on illegal housing, lack of electronic registration of previous housing, and other problems. In contrast, remote sensing technology can provide multi-resolution, multi-spectral, and multi-temporal image data in real time. Compared to other types of data, its significance is in terms of providing fine micro-level attribute qualities for real-estate studies. However, its application in real estate has gained more attention in recent times. Remote sensing images were blended with other raster images to reflect economic and social geographic information. This technique was used for fine-scale urban housing price mapping [3]. Another effective way of reflecting residential environmental conditions is to calculate the normalized difference vegetation index (NDVI) or use normalized spectral mixing analysis (NSMA) to calculate fractions of vegetation, and non-permeable surfaces and soil [4,5]. As the social and economic characteristics are determined in the dark [6,7], housing prices can be modeled [8,9]. Also, the airborne light detection and ranging (Li-DAR) technique measures the residential building footprint and volume [10], as well as residential values in coastal urban areas [11]. At the same time, remote sensing technology also has great potential in estimating the total volume of houses. For example, the total house volume can be estimated by extracting the height of the building and counting the number of floors, which is also the starting point of our approach.

There have been numerous studies conducted by researchers on building height calculation using remote sensing technology. In general, fine-scale building heights applicable to individual residential buildings can be estimated from three types of data [12]: (i) Li-DAR, (ii) radio detection and ranging (RADAR), and (iii) high-resolution optical images. However, LiDAR has a relatively high accuracy [13–15], and the coverage area of LiDAR is small. In particular, the acquisition cost of airborne LiDAR is high, while the measurement density of spaceborne LiDAR is not sufficient to cover all houses. RADAR also has certain possibilities [16–18]. At the same time, RADAR images usually record mixed signals from different microwave scattering mechanisms. Due to its geometry, there is a relatively high uncertainty in building height estimation [19]. High-resolution optical images can prevent this problem and can be widely used. There are two general ways to calculate building heights by high-resolution optical images: calculating building heights by using shadows [20–23], and using stereopairs to generate the digital surface models (DSMs) and extracting building heights from them [24,25]. The deep learning approach is also a new method in remote sensing that was recently considered by researchers who achieved estimation of building height with good accuracy [12].

In summary, remote sensing methods represent a new choice for total housing monitoring because conventional sampling survey methods in China are insufficient. Although there have been many studies and there are established methods available to estimate building height using remote sensing techniques, it is often insufficient to estimate the total amount of urban housing. To develop a more efficient and accurate mapping system for total housing estimation and calculate residential building heights compared to sample surveys, this study used Chinese satellite GF-7 stereopair images, point of interest (POI) data, and other data combined with both DSM and shadow methods. Based on this, the total value of housing was further calculated, and the driving factor model was derived to quantitatively analyze the REM development potential of Ningbo City. The innovations of this study lie in two aspects. Our engineering innovation was mainly to use the method of extracting building height to further calculate the total housing and per capita living area, which solves the bottleneck of statistical investigation. In addition, the analysis of residential potential is the main theoretical innovation of the study, including the derivation of the driving factor model.

The present study focused on the following main objectives: (i) estimation of total housing volume and per capita housing area in the urban Ningbo City area using remote sensing and other datasets; and (ii) assessing the potential for real-estate development in the Ningbo City urban area. The remainder of this article is organized as follows: Section 2 presents details of the study area, data, and the description of the methodological workflows. Further, Sections 3 and 4 present the results and discussion, respectively. Finally, conclusions are drawn in Section 5.

## 2. Materials and Method

### 2.1. Study Area Profile

The city of Ningbo is located in the middle part of the coastline of the mainland of eastern China. It is in the eastern part of Zhejiang Province, which is adjacent to Shanghai and Hangzhou. It is an economic center and an important port city in the southern branch of the Yangtze River Delta region along the southeast coast. It has been rated as the China's most promising city by the United Nations. As a modern port city with developed industry and commerce, Ningbo City has witnessed rapid economic development in recent times. The GDP of Ningbo City in 2021 reached CNY 1459.49 billion. The study area includes 6 urban districts of Haishu, Jiangbei, Zhenhai, Beilun, Yinzhou, and Fenghua (Figure 1).

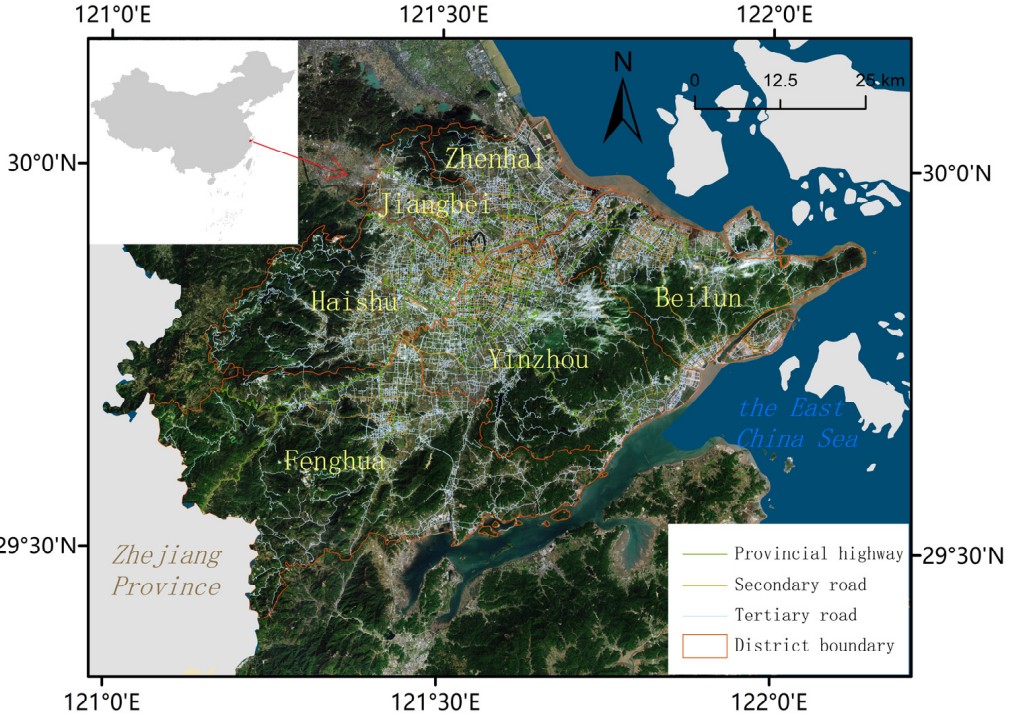

**Figure 1.** Location of study area and urban road network.

The population growth rate of urban areas in Ningbo City has remained at approximately 1.5 percent in the past 20 years due to the influx of migrants and rural residents into the city. With the development of the urban economy and people's demand for housing, real estate in the urban areas of Ningbo City has seen rapid development since 2016 (Figure 2). Due to policy and other factors, the volume of business decreased after 2018. However, housing prices still maintain an upward trend.

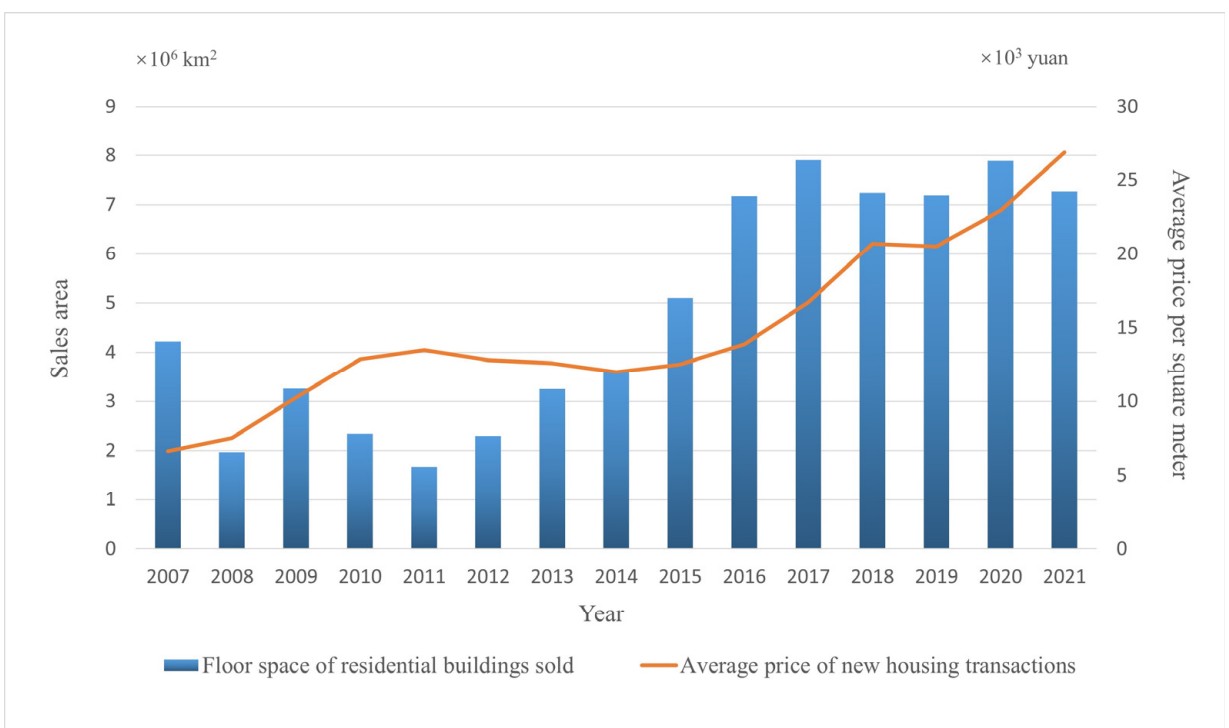

**Figure 2.** The trend chart of new house sales area and housing prices in Ningbo City from 2007 to 2021. Housing prices have maintained an upward trend (the data were obtained from the yearbooks of the Ningbo Bureau of Statistics).

### 2.2. Data Sources and Preprocessing

Three data sources were used in this study:

(1)   A total of 15 stereoscopic image pairs (Figure 3) composed of GF-7 front- and rear-view images (resolution: 0.8 m, panchromatic) of Ningbo urban area from December 2019 to January 2022 were used in this study. These were obtained from the high-resolution special project of the National Bureau of Statistics of China. For the areas not covered by GF-7 imagery, panchromatic-band remote sensing data of GF-1 and GF-2, as well as GF-6 of the same period, were used as a supplement with the shadow method to calculate the building height. The data were obtained from the Aerospace Information Research Institute, Chinese Academy of Sciences (http://ids.ceode.ac.cn/, accessed on 3 June 2023). The GF-1, GF-2 and GF-6 data were preprocessed by geometric correction. However, the GF-7 data are usually geometrically corrected after the DSM is created.

(2)   Building contour vector data, residential POI point data, and road vector data at all levels of Ningbo City were obtained from Amap (https://www.amap.com/, accessed on 3 June 2023) in 2022. Geospatial data including the remote sensing base map and the vector data of Ningbo City urban administrative division were also derived from the same archive as mentioned above.

(3)   The statistical data including population data, economic data, and residential sales data were taken from the Ningbo Municipal Statistical Yearbook. Finally, these data were used to calculate the PCLA and build the models.

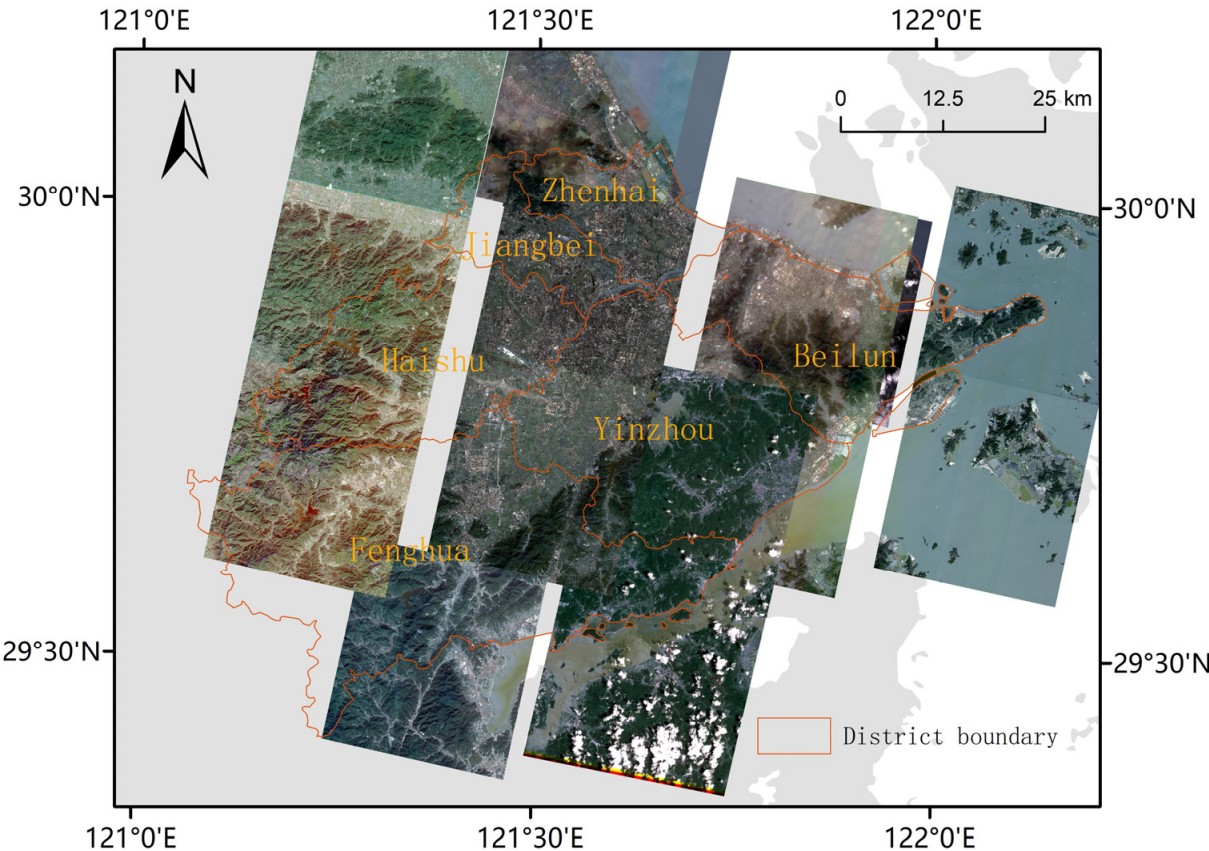

**Figure 3.** The urban coverage of GF-7 images. For the area that cannot be covered by GF-7 images, the shadow method was used to calculate the building height with GF-1, GF-2, and GF-6 data for the same period.

### 2.3. Research Framework

In this study, multi-source data such as high-resolution remote sensing images, POI, and statistical panels were used to obtain building heights and floor areas. They were used to estimate the total regional housing and per capita housing area, while forecasting the development potential of the REM. The research framework mainly included the following steps: the first step was to obtain the building height. Estimates of the total amount of housing were based on determining the height of the building. Considering the work efficiency and measurement accuracy, the height of the building was determined by the DSM method, which was supplemented by the shadow method. In the second step, the number of floors was calculated. The road network buffer zone was used to remove the underlying shops by counting the number of residential floors. In the third step, the number of floors was multiplied by the floor area to obtain the gross floor area and the total floor area. In the fourth step, the numbers of floors were aggregated to eliminate public areas. Subsequently, PCLA was calculated. Finally, real-estate development potential was analyzed according to the estimated PCLA and the results of the driving factor model. Figure 4 represents the workflow of this study.

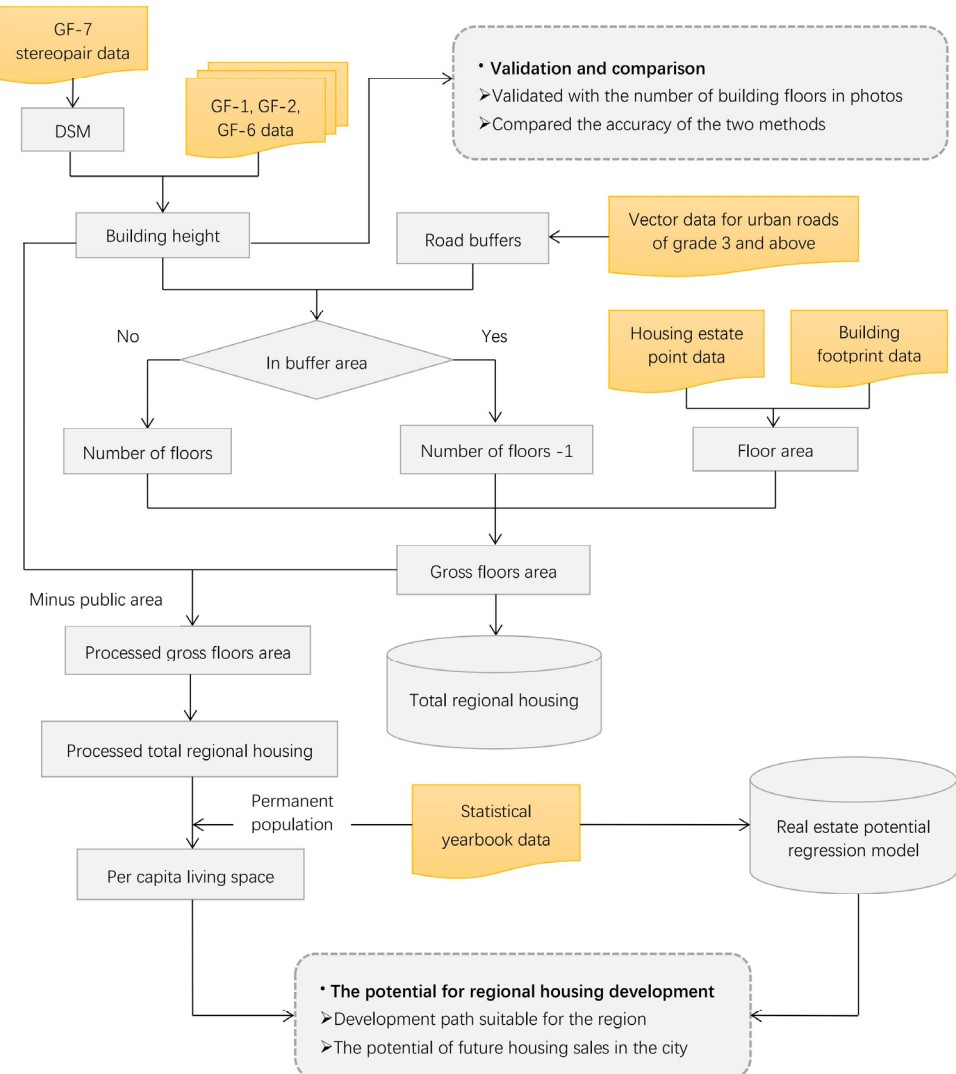

**Figure 4.** Total regional housing building estimation and development potential analysis workflow.

## *2.4. Research Methods*

### 2.4.1. Building Height Extraction Method

Building Height Estimation by the Digital Surface Model (DSM)

The digital surface model (DSM) is a ground elevation model that includes the height of ground objects such as buildings and trees. Unlike the digital elevation model (DEM), DSM also covers the elevation of surface information other than the terrain surface based on DEM, while DEM only contains terrain elevation information and does not contain other surface information. From the point of view of the data production process, the data source of the DSM has higher requirements for source resolution and product resolution. Therefore, it can play a big role in areas where the building height is required [24].

DSM can be generated from the GF-7 stereopairs, namely two front and rear panchromatic images. Figure 5 shows a group of stereopairs, the DSM, and the rendering details of Ningbo City urban area. The DSM data contains the elevation information of each pixel. The height information for the building can be obtained by subtracting the elevation of the road from the roof.

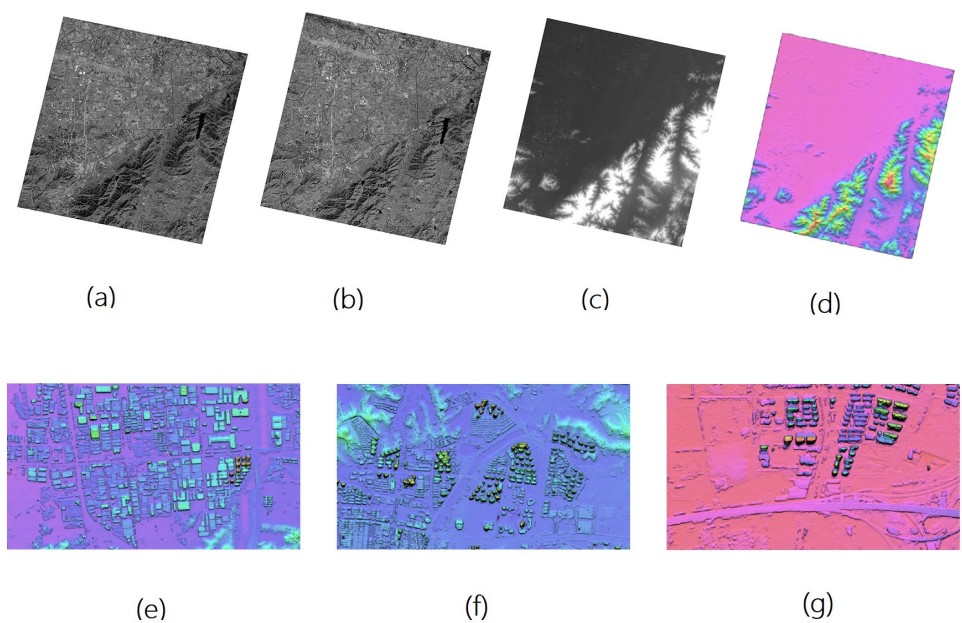

(a)　　　　　　　　(b)　　　　　　　　(c)　　　　　　　　(d)

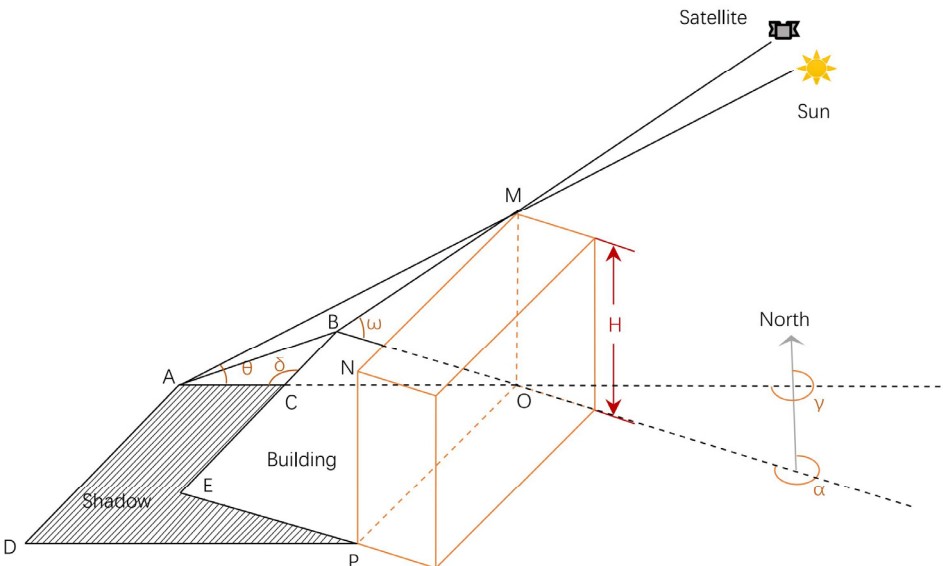

(e)　　　　　　　　　　　(f)　　　　　　　　　　　(g)

**Figure 5.** A set of GF-7 materials and DSM results from urban Ningbo City: (**a**,**b**) are the two front and rear panchromatic images; (**c**,**d**) are generated DSM and the DSM after rendering, respectively; (**e**–**g**) show details of the rendered DSM.

Estimation of Building Height by Shadows

A building height estimation method combining shadow length and a feature line segment [23] was developed. As is shown in Figure 6, $\omega$ is the satellite altitude angle, $\theta$ is the solar altitude angle, $\alpha$ is the satellite azimuth angle, $\gamma$ is the solar azimuth angle, and $\delta$ is the angle between the shadow line of the vertical intersection of the building and the strike line of the building in a clockwise direction (angle ACB).

**Figure 6.** Calculating building height by shadow method.

There are four ways to calculate the height of a building:

(1)　Calculate the height of the building using the full length of the shadow.

$$H = AO \times \tan\theta \tag{1}$$

(2)　Calculate the height of the building by the displacement of the corner point of the upper surface, which is caused by the difference in the height of the building.

$$H = BO \times \tan\omega \tag{2}$$

(3)　Calculate the height of the building using the connection length between the imaging point and the shadow point of the roof corner point.

$$\angle AOB = \alpha - \gamma \tag{3}$$

According to the law of cosines:

$$\begin{aligned} AB^2 &= AO^2 + BO^2 - 2 \cdot AO \cdot BO \cdot \cos(\alpha - \gamma) \\ &= H^2 \left( (\cot\theta)^2 + (\cot\omega)^2 - 2\cot\theta \cdot \cot\omega \cdot \cos(\alpha - \gamma) \right) \end{aligned} \tag{4}$$

Formula sorting:

$$H = \frac{AB}{\sqrt{(\cot\theta)^2 + (\cot\omega)^2 - 2\cot\theta \cdot \cot\omega \cdot \cos(\alpha - \gamma)}} \tag{5}$$

(4)　Calculate the height of the building using the visible shadow length under the building shield.

$$\delta = \angle CBO + \angle BOC = \angle CBO + \alpha - \gamma \tag{6}$$

Formula sorting:

$$\angle CBO = \delta - \alpha + \gamma \tag{7}$$

The sine theorem:

$$\frac{OC}{\sin\angle CBO} = \frac{BO}{\sin\angle BCO} \tag{8}$$

The building height can be obtained as:

$$H = \frac{AC \cdot \sin\delta}{\frac{\sin\delta}{\tan\theta} - \frac{\sin(\delta - \alpha + \gamma)}{\tan\theta}} \tag{9}$$

The first three methods are more commonly used because the 4th method measures $\delta$ and also produces measurement errors. From an experimental point of view, image measurements will produce random errors, which are significant in calculations. Therefore, when the solar elevation angle is small, the 1st and 3rd methods should be used to reduce the influence of random measurement error when selecting longer characteristic lines for measurement.

2.4.2. Estimation of the Total Housing Area in the Region

The total amount of housing was estimated by multiplying the floor area of the building by the number of floors and by calculating the total floor area of the statistical area. The number of floors of these residential buildings was reduced by one because of the city. Most of the shops were on the first floor of the building near the city road.

The main steps were:

(i) According to the point data of the residential area and the building contour vector data, the delimited residential buildings in the residential area and the area under a single building were calculated.

(ii) Buffer zones of a 30 m radius were created using urban roads above Grade 3 (Figure 1). Next, building height (*h*) was obtained using the DSM or shadow method. In this connection, we established 3 m as the standard for each floor because it is a common height for each floor of a residential building. For residential buildings inside and outside

the buffer zone (i.e., whether they intersect the buffer zone or not), two different calculation methods were used.

For residential buildings outside the buffer zone:

$$L = \left[\frac{h}{3}\right] \tag{10}$$

For residential buildings that overlap the buffer zone:

$$L = \left[\frac{h}{3}\right] - 1 \tag{11}$$

where [x] denotes the largest integer not greater than x.

(iii) The floor area $S$ was multiplied by the number of floors $L$ and then added to calculate the total housing area $M = \sum S \cdot L$.

### 2.4.3. Calculation of PCLA

The ratio of the residential public area to the total floor area is called the share ratio or share coefficient. For housing construction, Chinese regulations have clear standards. For example, for residential buildings, the share rate is 7 to 12 percent below 7 floors. For homes with 7 to 11 floors, the ratio is 10 to 16 percent and for flats with 12 to 33 floors, it is 14 to 24 percent. The share rate for villa-type housing is 1 to 8%. According to China's "Administrative Measures on Commercial Housing Sales", if the error ratio of the absolute area during the real-estate (sales) process is within 3%, the housing price payment will be settled according to the actual situation of payment. When the absolute area error ratio exceeds 3%, the buyer has the right to withdraw. Therefore, real-estate developers still have 3% wiggle room.

Based on the above considerations, the share coefficients to facilitate the integration calculations can be designed as below:

$$T = 0.05 \times \sqrt{L} \tag{12}$$

where $L$ represents the floors of the building.

The distribution area can be expressed by excluding the PCLA and the PCLA can be expressed as:

$$m = \frac{\sum S \cdot L \cdot (1 - T)}{R} \tag{13}$$

where $S$, $L$, $T$, and $R$ are the floor area, the number of floors, the share coefficient, and the total resident populations of the region, respectively.

### 2.4.4. The Derivation of the Driving Factor Model

People buying commercial housing can be roughly divided into two categories: (i) individuals with inelastic needs and (ii) individuals with advanced needs. The two different types of buyers reflect the two fundamental conflicts that derive from the development of the REM, i.e., the conflict between demand and supply and the economic conflict.

Supply and demand are related when considering the conflict between them. The number of people with inelastic needs largely depends on the total population and its structure. However, for a city like Ningbo with a large population influx and rapid population growth, the population growth rate (approximately 1.5%) is much higher than the natural growth rate (approximately 0.2%). Most of the inelastic demand in a year is generated by increased population flows. Assuming that the per capita housing area ($m$) in a year does not change significantly, the excess housing area ($y_1$) caused by inelastic demand, which is also the sales area of commercial housing ($Y$), is contributed by this part of the excess population ($r_b$). At this point, $Y = y_1 \approx m \cdot r_b$. Given that everyone may not

always be able to afford housing, some are acquired by renting space after being bought by others. Considering the driving factors, this is still caused by the conflict between supply and demand.

Since the per capita housing area can vary a certain amount each year, $m_1$ represents the current year's PCLA, $m_0$ represents the previous year's PCLA, $r$ is the permanent resident population of the previous year, $r_k$ is the permanent resident population of the current year, $r_b$ is the newly added population of the current year, and $z_1$ and $z_0$ are the total amount of real estate in the current year and the previous year, respectively. These are:

$$
\begin{aligned}
Y = z_1 - z_0 &= m_1 \cdot (r + r_b) - m_0 \cdot r \\
&= r \cdot (m_1 - m_0) + m_1 \cdot r_b \\
&= r_k \cdot (m_1 - m_0) + m_0 \cdot r_b
\end{aligned}
\tag{14}
$$

In the above formula, $m_0 \cdot r_b$ does not clearly represent the actual sales volume of inelastic demand when $m_0$ changes with time.

At this time, constant $\alpha$ is introduced.

$$
Y = r_k(m_1 - m_0) + \alpha \cdot r_b + (m_0 - \alpha) \cdot r_b
\tag{15}
$$

The $\alpha$ can be regarded as the expectation of new house sales area created by one with inelastic demand. It indicates the expectation of a per capita housing area with an additional population, which is relatively fixed in a period of years. Considering the term $(m_0 - \alpha) \cdot r_b$, since $m_0$ and $\alpha$ might be close in theory, the value of $r_b$ is relatively small; hence, it can be omitted as an error. We can obtain:

$$
Y = r_k(m_1 - m_0) + \alpha \cdot r_b
\tag{16}
$$

The $\alpha \cdot r_b$ is essentially the number of houses sold in a current year driven by inelastic demand. In other words, $r_k \cdot (m_1 - m_0)$ can be interpreted from economic conflicts. As the economy develops, people's economic power improves and they tend to buy improved housing. Therefore, the PCLA of the entire society can be improved. Social income is a pyramid structure where the number of high-income people is decreasing and the number of middle- and low-income people is increasing. For high-income earners, their demand for housing is not endless. They take into account both the specific economic environment and possible procurement restriction policies. Housing demand does not vary significantly with individual economic conditions. As the overall social economy develops, middle- and low-income groups improve. With disposable income rising relative to the housing prices, the number of potential house buyers has improved in an exponential pattern. This social and economic development is also reflected in the increase in housing space per capita. Considering the disposable income per capita in the society as $s$, and the average housing price as $j$, we can obtain $r_k(m_1 - m_0) = \beta \cdot r_k (\exp(s/j) - 1)$, which reflects the economic environment driven by commercial housing sales.

Let $y_1$ and $y_2$ be the annual commercial housing sales driven by inelastic demand and economy, respectively. The gross sales amount considered as housing sales will be:

$$
Y = y_1 + y_2 = \alpha \cdot r_b + \beta \cdot r_k \left( e^{\frac{s}{j}} - 1 \right)
\tag{17}
$$

where $r_b$, $r_k$ and $s$ are the newly increased permanent resident population, total permanent resident population, and per capita disposable income of the city in the current year, respectively. The $j$ is the average price of commercial housing in the city, which can be obtained by dividing the total urban sales by the total urban sales area. Here, $\alpha$ and $\beta$ are regression coefficients.

There are two reasons for omitting the per capita housing area ($m$) in the derivation. First, regional economic data are often more predictable than the per capita housing area. All statistics were taken from official statistical yearbooks to ensure relative authority and

accuracy. However, the per capita housing area in the statistical yearbook was obtained by sample survey [2] and its accuracy is not sufficient for regression modeling.

## 3. Results

### 3.1. Checking the Accuracy of Extraction of Building Height

The DSM and shadow methods were used to test 10 different buildings and the results were compared (Table 1). Since the main objective was to determine the number of floors according to the height of the building, the number of floors in the field photo of the building was used as the inspection standard. As is presented in Table 1, building floors were estimated using the two methods. The errors between the predicted number of floors and the actual number of building floors were controlled to one floor, indicating that these methods can meet the requirements of calculation accuracy.

**Table 1.** Comparison of building heights and floors estimated by two methods.

| Building Number | GF-7 DSM Estimated Height (m) | The Number of Floors Predicted by GF-7 DSM | Height Measurement by Shadow Method (m) | The Number of Floors Predicted by the Shadow Method | Photo Floor Number |
|---|---|---|---|---|---|
| 1 | 79.9 | 26 | 77.6 | 25 | 25 |
| 2 | 75.7 | 25 | 77.2 | 25 | 25 |
| 3 | 15.1 | 5 | 15.1 | 5 | 5 |
| 4 | 32.7 | 10 | 29.6 | 9 | 10 |
| 5 | 17.7 | 5 | 18.3 | 6 | 6 |
| 6 | 32.5 | 10 | 30.8 | 10 | 11 |
| 7 | 14.6 | 4 | 13.9 | 4 | 4 |
| 8 | 33.4 | 11 | 31.6 | 10 | 11 |
| 9 | 55.6 | 18 | 51.5 | 17 | 17 |
| 10 | 75.7 | 25 | 72.1 | 24 | 25 |

### 3.2. Total Housing Area and per Capita Housing Area

As is shown in Figure 7, most of the buildings are below 30 m and Haishu and Yinzhou had the highest number of residential buildings. By calculating and summarizing the total housing area of the six districts of Ningbo City, the total housing area of the urban area in 2020 was found to be approximately 152.8 km$^2$. The urban districts in descending order of total housing area are Yinzhou, Haishu, Beilun, Jiangbei, Zhenhai, and Fenghua (Figure 8).

The per capita housing area of each district in Ningbo City also varies significantly (Figure 8). Haishu, Jiangbei, and Yinzhou are the core areas of Ningbo City, occupying most of the urban flat area of Ningbo City. The real-estate industry has experienced long-term and robust growth over the past few decades, driven by the economy and demographics. Consequently, the per capita housing area is also high and reaches 30.3 m$^2$, 28.2 m$^2$, and 26.9 m$^2$ in Haishu, Jiangbei, and Yinzhou, respectively. The per capita housing living area of Fenghua District is relatively lower. Due to the district's mountainous and restricted plains, regional real estate has not been developed to its full potential. As Fenghua District was not included in the Ningbo City urban area before 2016, the economy and population were relatively backward, and real estate was not developed. The per capita housing area of Fenghua District is the lowest at 20.6 m$^2$. Beilun District and Zhenhai District are located along the coast. To save transportation costs and improve economic benefits, there are a large number of processing plants in the coastal area of Ningbo City. Due to the limited availability of residential land in these two districts, the per capita housing area of the regions was affected. The sizes were measured at 22.9 m$^2$ and 23.9 m$^2$, respectively.

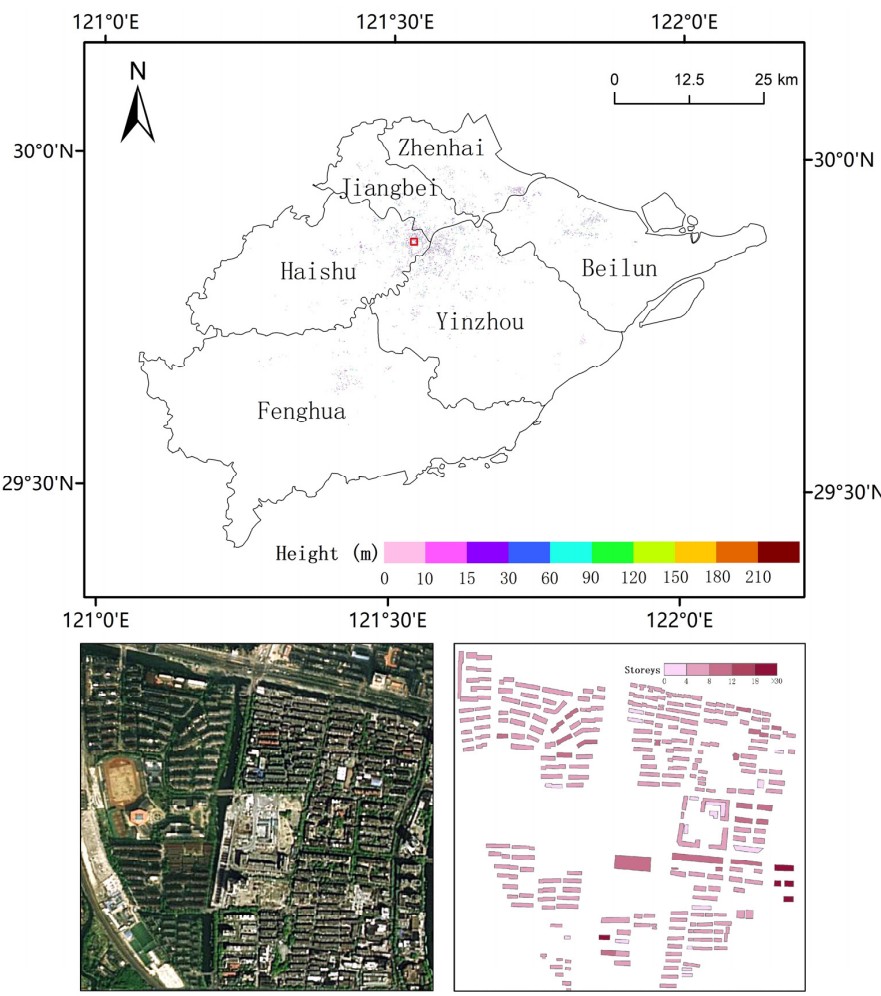

**Figure 7.** Height and spatial distribution of residential buildings. The architectural details of the residential buildings in the area within the red rectangle are shown.

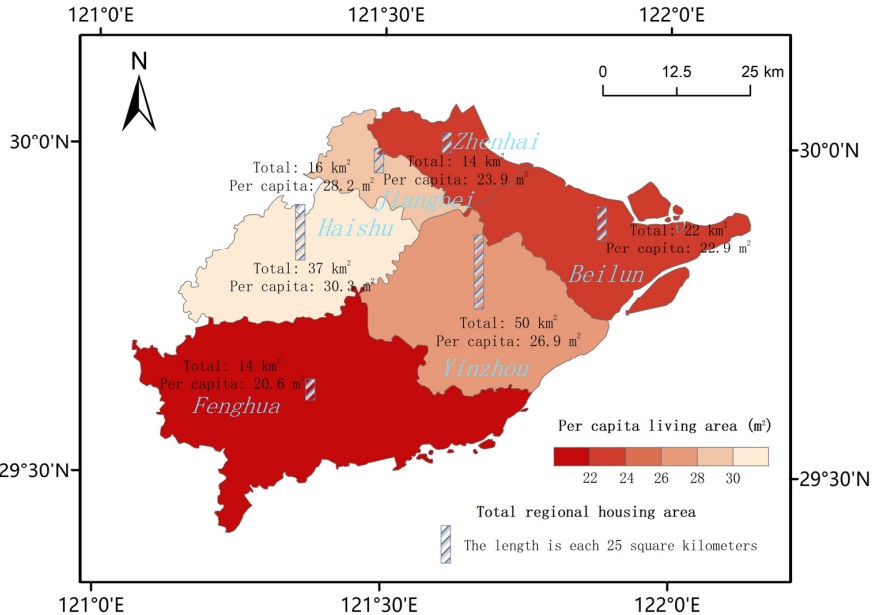

**Figure 8.** The total housing area and per capita housing area of each district in Ningbo City. The per capita housing area varies significantly by region.

### 3.3. Analysis of Driving Factor Model Calculations

Regression modeling was conducted using Equation (17) and relevant data were used from a series of Statistical Yearbooks from 2007 to 2021. Since the statistical yearbooks did not contain the permanent population (the actual long-term resident population), the registered population (the population belonging to the official register of the region) was used. The meaning of $\alpha$ was changed when using the registered population. It no longer represents the expectations of housing area per capita driven by inelastic demand. But there is a proportional relationship between the two. The proportion is almost equal to the ratio of the newly registered population to the new permanent resident population.

The regression results are $\alpha = 118.34$, $\beta = 0.019$, and the coefficient of determination $r^2 = 0.93$. Figure 9a shows the scatter-fit plot of the model.

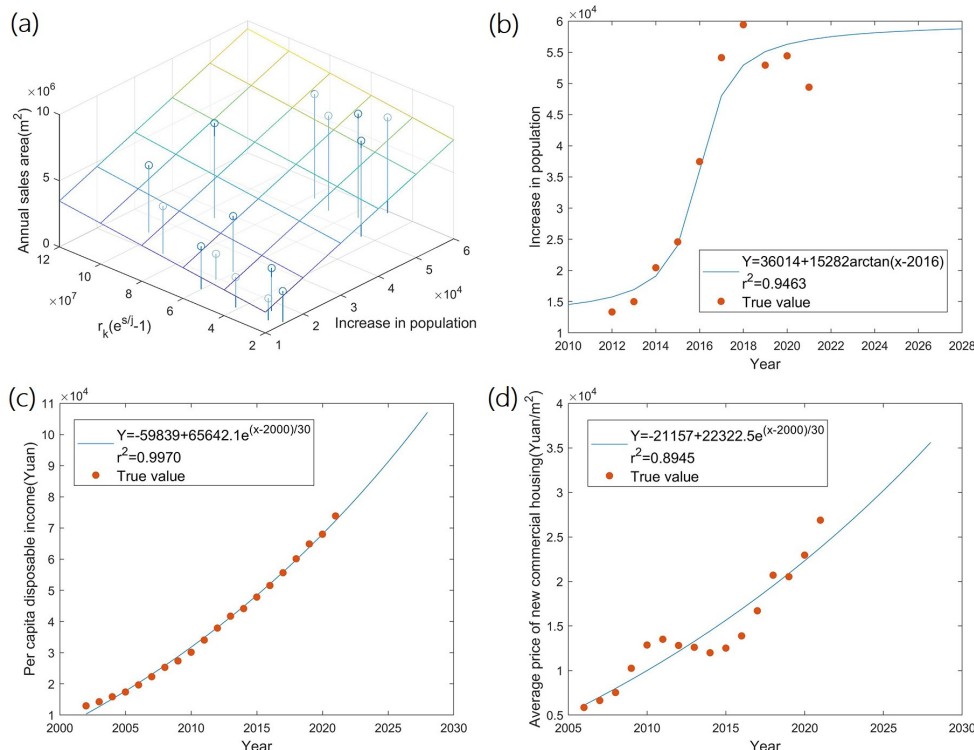

**Figure 9.** (**a**) shows the scatter-fit plot of the model, (**b**–**d**) are the fitting functions of additional population, per capita disposable income, and housing price change over time, respectively.

The data presented in Table 2 were obtained by sorting the yearbook data and model calculation results, which show the proportion of inelastic demand and the improved demand area and sales area in the Ningbo City urban area from 2007 to 2021. The sum of the two is the predicted total sales volume of the year. Actual sales volumes are listed for comparison.

The predicted results of the model (Table 2) are relatively close to the actual value of the statistical yearbook, indicating that the driving factor model has good predictive ability. There were some errors in 2016 because the administrative division of Ningbo City has undergone significant changes. The Jiangdong District (the municipal government's administrative center) was incorporated into Yinzhou District, and a part of Yinzhou District was divided into Haishu District. Fenghua City was also changed into Fenghua District and included in the Ningbo City urban area. In addition to the changes in the total area of the district and the population of each district, the house prices in Yinzhou District in one year increased by approximately 50%, which is still the highest among all districts. The biggest flaw was in 2008 when China's economic growth, driven by external demand, slowed sharply as a result of the global economic slowdown following the subprime crisis. This has affected the willingness of residents to buy houses. With the development of the

national rescue system, sales of urban housing in Ningbo City reached 3,263,321 square meters in 2009.

**Table 2.** Comparison of the predicted housing sales area by the driving factor model and the real sales area.

| Year | Predicted Housing Sales Area of Inelastic Demand (m²) | The Percentage of Inelastic Demand Sales in Total Sales | Predicted Housing Sales Area of Improved Demand (m²) | The Percentage of Improved Demand Sales in Total Sales | Predicted Total Sales Area of the Year (m²) | The Real Sales Area of the Year (m²) |
|------|------|------|------|------|------|------|
| 2007 | 2,808,875 | 70.86% | 1,155,050 | 29.14% | 3,963,925 | 4,223,107 |
| 2008 | 2,294,061 | 66.37% | 1,162,386 | 33.63% | 3,456,447 | 1,966,489 |
| 2009 | 2,013,694 | 78.05% | 566,454 | 21.95% | 2,580,148 | 3,263,321 |
| 2010 | 1,797,709 | 81.78% | 400,569 | 18.22% | 2,198,278 | 2,343,976 |
| 2011 | 1,647,998 | 77.12% | 489,027 | 22.88% | 2,137,025 | 1,673,491 |
| 2012 | 1,625,867 | 67.45% | 784,593 | 32.55% | 2,410,460 | 2,299,346 |
| 2013 | 1,754,511 | 60.58% | 1,141,859 | 39.42% | 2,896,370 | 3,253,458 |
| 2014 | 2,419,155 | 58.89% | 1,688,861 | 41.11% | 4,108,016 | 3,622,723 |
| 2015 | 2,948,644 | 59.89% | 1,974,705 | 40.11% | 4,923,349 | 5,098,873 |
| 2016 | 4,390,362 | 71.02% | 1,791,935 | 28.98% | 6,182,297 | 7,177,824 |
| 2017 | 6,488,438 | 81.37% | 1,485,370 | 18.63% | 7,973,808 | 7,901,227 |
| 2018 | 7,031,538 | 87.87% | 970,215 | 12.13% | 8,001,752 | 7,243,089 |
| 2019 | 6,263,576 | 82.91% | 1,291,091 | 17.09% | 7,554,667 | 7,196,155 |
| 2020 | 6,440,980 | 85.77% | 1,069,043 | 14.23% | 7,510,023 | 7,892,762 |
| 2021 | 5,845,452 | 87.10% | 865,365 | 12.90% | 6,710,817 | 7,265,444 |

With the economic and social development of Ningbo City in recent years, a large number of non-local people have moved in and settled. Therefore, the proportion of inelastic demand driving commercial housing sales has been rising in recent years (Table 2). Ningbo City's REM is in a strong position to reach its full potential in the near future thanks to strong and persistent demand. The ratio driven by improved demand is mainly influenced by the ratio between per capita disposable income and housing prices. When income growth fails to keep pace with the housing price growth, the growth in advanced housing slows and the ratio declines. As Ningbo City's overall economy is strong, advanced housing can also sustain a large sale. Therefore, Ningbo City's REM still has significant room for long-term development.

*3.4. Real-Estate-Potential Prediction*

Using regression models of driving factors, we can make a reasonable forecast of housing sales over the next few years. Since the model's prediction require the use of data on excess population, per capita disposable income, and the average price of commercial housing, these data are first fitted to obtain the data for the next few years.

Accurately predicting overpopulation as a variable is more challenging. The overpopulation in Ningbo City is mainly from other places, which is not only related to local factors, but also influenced by the external population and economic factors. Considering the characteristics presented by additional population data, this presented phase changes under the influence of the regional economic environment. It exhibits a sudden increase or decrease over a period of time and then flattens out. In today's global economic environment, China's total import and export volume also maintained growth in 2022. As a port city, it is advantageous for Ningbo to enhance and consolidate its economic development. It can be predicted that Ningbo will attract immigrants in the next few years. Therefore, additional population data from 2012 to 2021 in the Ningbo Statistical Yearbook was used for fitting. At this stage, the new population growth changes from slow growth to rapid growth and finally to a steady level. As is shown in Figure 9b, a conservative and optimistic

estimate was made for the overpopulation of the Ningbo City urban area in the next few years, which has been stable in recent years.

Per capita disposable income is relatively easy to predict, which has been steadily increasing with economic development. In particular, Ningbo City itself is a foreign trade port, which makes its economy more likely to grow in the next few years and drive steady growth in per capita disposable income. This study compared urban per capita disposable income from 2002 to 2021 in the Statistical Yearbook. As is shown in Figure 9c, urban per capita disposable income in Ningbo City has improved rapidly due to GDP growth.

Housing price changes fluctuate little in the short term, but in the long run, this is more consistent with the simultaneous increase in income. Especially in first-tier cities of a strong economy like Ningbo City, housing prices will tend to increase in the future. In this paper, the new commercial housing prices in the Ningbo City urban area from 2006 to 2021 (obtained by dividing the total sales volume of commercial housing in urban areas by the sales area) was fitted (Figure 9d).

The fitting function was used to predict the new population, per capita disposable income and housing prices over the next 5 years (2023 to 2027). The urban population was obtained by adding the urban population in 2021 and the additional population every year thereafter. By substituting the above data into the driving factor model, the predicted sales area of residential commercial housing in 2023–2027 was calculated. The details of residential commercial housing sales forecast data are presented in Table 3.

**Table 3.** 2023–2027 residential commercial housing sales forecast.

| Year | Predicted Additional Population | Predicted Per Capita Disposable Income (Yuan) | Predicted Average Price of Commercial Housing (Yuan/m$^2$) | Predicted Urban Population | Predicted Housing Sales Area of Inelastic Demand (m$^2$) | Predicted Housing Sales Area of Improved Demand (m$^2$) | Predicted Total Sales Area of the Year (m$^2$) |
|---|---|---|---|---|---|---|---|
| 2023 | 57,851 | 81,461 | 26,894 | 3,227,791 | 6,846,087 | 1,206,680 | 8,052,767 |
| 2024 | 58,119 | 86,251 | 28,523 | 3,285,910 | 6,877,802 | 1,221,899 | 8,099,701 |
| 2025 | 58,328 | 91,202 | 30,207 | 3,344,238 | 6,902,535 | 1,237,490 | 8,140,025 |
| 2026 | 58,496 | 96,322 | 31,947 | 3,402,734 | 6,922,416 | 1,253,617 | 8,176,033 |
| 2027 | 58,634 | 101,615 | 33,747 | 3,461,368 | 6,938,747 | 1,269,900 | 8,208,648 |
| Total | 291,428 | - | - | - | 34,487,589 | 6,187,186 | 40,677,174 |

Table 3 presents projections of commercial residential sales over the next five years with a total of 40,677,174 square meters. A single year's sales forecast may be affected by actual data for that year. Therefore, the total annual sales volume may fluctuate. However, the total volume may be relatively stable within five years. In the next five years, the demand for commercial housing due to population influx is expected to reach 34,487,589 square meters, which shows that the short-term demand of the Ningbo City urban REM is still substantial. The market is likely to be tapped for at least next five years. A large amount of inelastic demand can create greater potential for the REM, provided that regional economic development attracts an influx in terms of population.

## 4. Discussion

### 4.1. Comparison with Previous Studies and Traditional Methods

Building height measurement methods that have been studied in the past include building height estimation by DSM [26,27], building height estimation by shadow [20–23], and the method of deep learning [12]. The DSM is a widely used method for estimating building height. In some studies, building height is calculated by subtracting the DEM from DSM, and the accuracy of this method depends on high-precision DEM. And other studies use morphological top-hat by reconstructing the DSM to calculate normalized DSM (nDSM) to represent the relative height above the ground [26]. In general, the building height

accuracy of these methods is limited. Due to the high precision requirement of building height for accurate calculation of total housing, we directly calculated building height by subtracting the ground height from the top floor height of the DSM. Although the workload is increased, the accuracy is greatly improved. In addition, compared with previous methods, the high-resolution GF-7 image data ensures the accuracy of building height calculation with both DSM and shadow methods, which highlights the potential of GF-7 application in urban research. Another deep-learning-based building height estimation method [12] uses ZY-3 satellite images as multi- view training data. Its resolution is no higher than 2.1 m, but the root mean square error (RMSE) for the output building height results is only approximately 6 m. Comparing the building heights in Table 1, the average error of our DSM method using GF-7 images is likely to be within 3 m. Since GF-7 images have a higher resolution of 0.8 m, it can be considered that the accuracy of the deep learning method is close to that of the DSM methods. However, based on the multi-view images used, it can be assumed that the learning of the neural network in that study is still mainly based on the mathematical principles of generating the DSM from multi-view images. We therefore believe it to be no more accurate than the DSM.

Although these methods are widely used to estimate building height, they are less frequently used to monitor the total amount of housing in large areas. Based on these techniques and considering the conditions of the study area, this study developed a scientific method system for calculating the total amount of regional housing and per capita housing area. Considering China, where widespread and comprehensive electronic housing registration has not yet been implemented, this approach offers greater efficiency, less manpower input, and a shorter time frame than the statistical methods of sampling surveys [2]. The accuracy of this method is more than 90% when only considering the number of floors calculation. Theoretically, sampling surveys will not be as accurate as the method in this study, since it is difficult to cover most samples. However, our method successfully covered the residential buildings in the study area. According to the Statistical Yearbook of Ningbo, the urban per capita private housing area of residents was 47.11 $m^2$ in 2020, which was the per capita area of the registered population. The total urban housing area was found to be about 144.3 square kilometers, which was lower than the 152.8 square kilometers obtained in this study. The lack of statistics on housing owned by the non-resident population is also one of the reasons for the lower figure.

*4.2. Policy Implication*

Considering the different development statuses of real estate in each district of Ningbo City, each district should adopt a different path according to regional characteristics. Haishu, Jiangbei, and Yinzhou are considered to be the core areas that are densely populated and economically developed. As the supply of land is relatively limited, land and housing prices have increased. It is suitable for developing high-end housing for people who need improvement. With low housing prices, Fenghua District can provide affordable housing needs to reduce housing pressure due to low per capita housing area. As a coastal area, Beilun District has a long coastline, which is suitable for building resorts and houses with a sea view, while attracting certain people to purchase.

From a regional and national perspective, macro-level policy coordination is needed to maximize the REM potential. Large credit risk and high leverage ratios are common problems in the current world economy. China's financing system is mainly based on indirect financing in the form of bank loans, while in Europe and America, corporate financing is mainly responsible for financing in the form of equity due to the development of an open capital market. Bankruptcy of real-estate companies due to high debt has affected the confidence of China's REM and could lead to a depression in 2022. Hence, financial reforms to promote direct financing in the real-estate market will reduce corporate leverage and credit risks.

From the perspective of development trends, the long-term existence of "land finance" binds economic growth; the value of housing and the value of land is tied to finance,

and this results in high housing prices. This has not only become a strong obstacle in solving the housing problem of the residents of big cities, but has also seriously restricted residents' consumption. This has led to wealth polarization, which has hindered the further development of real estate. Moreover, the marginal benefit of investment is declining at an accelerating rate in the current stock economy. With the decline in China's population and the advancement of artificial intelligence (AI) and other technologies in the future, the industrial employment capacity will weaken, and it will become more dependent on the development of the service industry. Therefore, the economic development model dependent on "land finance" is not sustainable. Since the development of the service industry depends on urbanization and a high population density, only by completing the transition from the production and investment mode to the service mode can the government absorb the population and promote the development of the REM, while maintaining economic growth.

*4.3. Contributions and Limitations*

The contribution of our research is mainly divided into three aspects:

(1) This study estimated the total housing and PCLA of Ningbo City. It quantitatively analyzed future real-estate development potential. The results can provide reference data for urban development research and policy planning in Ningbo City. The research methods can also be applied to other cities of the same type.

(2) The study provided a more efficient and accurate method for total housing estimation compared to the sample survey [2]. It can play a direct role in real-estate field studies and be used to analyze housing characteristics in more detailed sub-regions. This can provide greater flexibility when combined with other geospatial data analysis.

(3) The driving factor model derived from the research is also a new method for analyzing real-estate potential. In addition to predicting the REM potential, the model can calculate the total amount and proportion of inelastic demand and improvements, which are difficult for surveys to accurately reflect. It has definite potential for other research related to urban development and real estate. The model also has the advantage of using fewer parameters to avoid a series of problems caused by less data.

However, the limitation is the fact that sudden changes in policy or the economic environment cannot be immediately reflected by the driving factor model. Therefore, it is necessary to collect data and recalculate parameters under this environment.

**5. Conclusions**

In this study, a methodology system based on high-resolution stereopair remote sensing was established to calculate the total amount of regional housing and per capita housing area. Compared to traditional sample survey methods, this method is more efficient, more accurate, and more flexible in terms of analysis. Based on the Chinese satellite GF-7 DSM, supplemented by other data, the total housing area and per capita housing area of the Ningbo City urban area were estimated. The total urban housing reached 152.8 km$^2$. This study has analyzed the real-estate development path suitable for each district. In addition, it developed a multiple regression model of real-estate development based on the driving factors of house purchases. It also analyzed the development potential of the Ningbo City REM through driving factors and changes in predicted sales volume ratios. It is predicted that the real-estate sales volume may reach 40 km$^2$ in the next five years. Finally, it can be concluded that Ningbo City still has considerable potential for real-estate development.

**Author Contributions:** Conceptualization, methodology, data curation, formal analysis, visualization, writing—original draft preparation, X.D.; conceptualization, methodology, project administration, L.W.; writing—review and editing, F.T., S.X., S.M. and B.N.; supervision, Z.N. All authors have read and agreed to the published version of the manuscript.

**Funding:** This study was supported by the National Key Research and Development Program of China (2022YFB3903702), and the High-Resolution Earth Observation System Project (20-Y30F10-9001-20/22).

**Data Availability Statement:** Not applicable.

**Acknowledgments:** We appreciate the data support provided by the National Bureau of Statistics. We would like to express our sincere thanks to the anonymous reviewers for their valuable comments and suggestions that helped us to modify the manuscript in its final form.

**Conflicts of Interest:** The authors declare that they have no known competing financial interest or personal relationships that could have appeared to influence the work reported in this paper.

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
