# Peer review of "Estimation and Development-Potential Analysis of Regional Housing in Ningbo City Based on High-Resolution Stereo Remote Sensing"

_remotesensing, doi:10.3390/rs15163953_

Round 1

Reviewer 1 Report

This study uses stereo pairs to obtain height information of urban buildings, estimate residential area, analyze the scale of new housing demand, and evaluate the potential for urban housing development in Ningbo. The research has good application value. However, there are still some shortcomings.

1. The author mentioned in the article that there have been studies using stereo pairs to extract building heights. What are the innovative points of this study compared to previous research content. Is the analysis of residential potential an innovation in this study, which the author did not emphasize in the introduction.

2. Figures 1 and 8 are not clear enough.

3. Lines 89-90, "This study used GF-7 stereopairs, POI data and other data combined  with both DSM and shadow methods to calculate residential building heights”. The author needs to clarify whether this study calculates the building height or the total amount of urban housing.

4. Lines 131-132. “However, the GF-7 data are  usually geometrically corrected after or during the generation of DSM.”  Is geometric correction performed before or after the production of DSM in this study?

5. POI data including building control vector data, residential POI point data and 133 road vector data at all levels of Ningbo city were observed from Amap 134( https://www.amap.com/ )In 2022 . Using “and” instead of “including” is more accurate.

6. It is mentioned in Figure 4 that this study used highways to create buffer zones, which have a low density and extremely limited coverage, making this method incorrect. Please clarify.

7. "The height of  the building can be observed by subtracting the elevation of the road from that of the road."  Please provide more detailed information. How to obtain the height of buildings without roads nearby?

8. Please modify the content of the discussion section, as the differences and similarities between the specific results of this study and previous studies have not been discussed.

Extensive editing of English language required

Reviewer 2 Report

The use of stereo remote sensing images to quantitatively analyze urban real estate holdings and development potential is a very novel idea and has been experimentally verified to be feasible, providing a valuable reference for assessing the current status of regional real estate development and development potential on a medium to large scale. This paper provides readers with a very interesting perspective to measure the development trend of real estate market. The only two issues are (1) the language presentation needs to be slightly modified to allow the reader to better understand the research methodology used and the analysis of the experimental results, and (2) the analysis results combined with the POI data are more applicable to the residential real estate market and seem to be less likely to yield desirable conclusions for commercial properties.

The use of stereo remote sensing images to quantitatively analyze urban real estate holdings and development potential is a very novel idea and has been experimentally verified to be feasible, providing a valuable reference for assessing the current status of regional real estate development and development potential on a medium to large scale. This paper provides readers with a very interesting perspective to measure the development trend of real estate market. The only two issues are (1) the language presentation needs to be slightly modified to allow the reader to better understand the research methodology used and the analysis of the experimental results, and (2) the analysis results combined with the POI data are more applicable to the residential real estate market and seem to be less likely to yield desirable conclusions for commercial properties.

Reviewer 3 Report

Authors used GF-7 stereopairs, POI (Point of Interest) data and 19 other data combined with digital surface model and shadow methods to calculate the height of residential buildings. The method is highly exploratory, but there are still several issues

(1) The various roads in the legend of Figure 1 are not clearly visible, it is recommended to depict them more clearly

(2) The DSM method used in the paper can indeed obtain height information of buildings by removing DEM data. However, to my knowledge, the resolution of DSM data obtained using GF-7 data is about 3-5 meters. So, what is the resolution of DEM used by the author in the paper? What is the vertical resolution of the DSM obtained from GF-7 data? In Table 1, the author displays several building height information and finds that it is not significantly different from the actual information. Can all buildings meet this accuracy, or can this effect only be achieved after reaching a certain level of height?

(3) What is the accuracy of using shadows to calculate the height of buildings in the paper? When the sun is at a certain height, not all shadows of buildings can be recognized by the image on the surface. Some shadows of higher buildings may appear on adjacent building objects. How can we calculate the height of such buildings? At the same time, the dense distribution of buildings in the Ningbo area, with differences in building methods and forms, may also lead to the phenomenon of overlapping shadows. How does the author consider these factors?

(4) How is the contour of the building in Figure 7 extracted? What type of image is used? What is the extraction method used?

(5) How did the author fit the b and d curves in Figure 9? I feel that fitting the b curve as a straight line is also possible.

The Quality of English Language is well,Language is easy for readers to read and understand.

Round 2

Reviewer 1 Report

The revised manuscript made unclear changes and contained some basic errors, such as spelling mistakes. The author did not diligently make revisions or address the reviewer's concerns. Specifically, the following issues were identified:

1.The author should emphasize the unique features of the research in the introduction rather than the discussion section.

2.The image quality did not improve.

3.In the response, the author claimed to have modified the road in the image, but no actual changes were made.

4.The discussion section was not revised to include a comparative analysis of the current study with existing research.

Additionally, the projection of the Chinese region in the location diagram is incorrect, and the South China Sea area is missing.

Reviewer 3 Report

All the issues I mentioned have been resolved. It is recommended to verify this method elsewhere

Further enhance the English description of the method
